# A Seroprevalence Study of Anti-SARS-CoV-2 Antibodies in Patients with Inflammatory Bowel Disease during the Second Wave of the COVID-19 Pandemic in Italy

**DOI:** 10.3390/medicina57101048

**Published:** 2021-10-01

**Authors:** Mirko Di Ruscio, Gianluigi Lunardi, Dora Buonfrate, Federico Gobbi, Giulia Bertoli, Donatella Piccoli, Antonio Conti, Andrea Geccherle, Angela Variola

**Affiliations:** 1IBD Unit, IRCCS Sacro Cuore Don Calabria Hospital, 37024 Negrar di Valpolicella (VR), Italy; andrea.geccherle@sacrocuore.it (A.G.); angela.variola@sacrocuore.it (A.V.); 2Medical Analysis Laboratory, IRCCS Sacro Cuore Don Calabria Hospital, 37024 Negrar di Valpolicella (VR), Italy; gianluigi.lunardi@sacrocuore.it (G.L.); donatella.piccoli@sacrocuore.it (D.P.); antonio.conti@sacrocuore.it (A.C.); 3Department of Infectious, Tropical Diseases and Microbiology, IRCCS Sacro Cuore Don Calabria Hospital, 37024 Negrar di Valpolicella (VR), Italy; dora.buonfrate@sacrocuore.it (D.B.); federico.gobbi@sacrocuore.it (F.G.); giulia.bertoli@sacrocuore.it (G.B.)

**Keywords:** seroprevalence, anti-SARS-CoV-2 antibodies, inflammatory bowel disease, COVID-19

## Abstract

*Background and Objectives:* Studies have shown a lower prevalence of anti-SARS-CoV-2 antibodies in patients with inflammatory bowel disease (IBD), including amongst those receiving biological therapy. Aims were to determine the seroprevalence of anti-SARS-CoV-2 antibodies in IBD patients and to assess any association between seropositivity and IBD characteristics. *Materials and Methods:* Serum from adult IBD patients was prospectively collected between December 2020 and January 2021 and analyzed for anti-SARS-CoV-2 antibodies. Information about IBD characteristics and SARS-CoV-2 exposure risk factors was collected and analyzed. Serum from non-IBD healthcare workers formed the control group. *Results:* 311 IBD patients on biologics and 75 on mesalazine were enrolled. Ulcerative colitis (UC) extension (*p* < 0.001), Crohn’s disease (CD) phenotype (*p* = 0.009) and use of concomitant corticosteroids (*p* < 0.001) were significantly different between the two IBD groups. Overall seroprevalence among IBD patients was 10.4%. The control group showed a prevalence of 13.0%, not significantly different to that of IBD patients (*p* = 0.145). Only a close contact with SARS-CoV-2 positive individuals and the use of non-FFP2 masks were independently associated with a higher likelihood of seropositivity amongst IBD patients. *Conclusion:* In IBD patients, the prevalence of anti-SARS-CoV-2 antibodies is not determined by their ongoing treatment. Disease-related characteristics are not associated with a greater risk of antibody seropositivity.

## 1. Introduction

The ongoing COVID-19 pandemic, caused by severe acute respiratory syndrome coronavirus 2 (SARS-CoV-2), has had a significant global impact [1].

Certain categories of people, including those with chronic inflammatory conditions, are more vulnerable to developing COVID-19 and suffering severe clinical outcomes [2].

Many patients with inflammatory bowel disease (IBD), including ulcerative colitis (UC), Crohn’s disease (CD) and inflammatory bowel disease unclassified (IBDU), require immunosuppressant medication that is associated with an increased risk of serious and opportunistic infections [3,4]. However, during the initial phase of the SARS-CoV-2 outbreak, scientific societies and international organizations did not recommend halting the use of immunomodulators or biologics due to the serious and potentially lethal consequences of discontinuing treatment [5,6,7].

Studies carried out thereafter have shown that IBD patients do not suffer a more severe or complicated form of COVID-19 than that of the general population, and that the rate of SARS-CoV-2 infection amongst IBD patients is comparable to that of the general population [8,9,10].

However, as diagnostic nasopharyngeal swabs were initially only carried out in symptomatic cases, many asymptomatic individuals may have been missed, and the true SARS-CoV-2 infection rate amongst IBD patients and amongst the general population is likely to be underestimated [11].

Serological testing to detect anti-SARS-CoV-2 antibodies has proven vital for epidemiological surveys on SARS-CoV-2 infection and in identifying asymptomatic cases [12,13].

Relatively few studies have investigated the prevalence of SARS-CoV-2 serum antibodies in IBD patients, and of these studies contrasting results have been reported. As yet, most studies have addressed the first phase of the COVID-19 pandemic only [14,15,16,17,18,19].

Thus, the aims of this study are to assess the seroprevalence of anti-SARS-CoV-2 antibodies among IBD patients during the second wave of SARS-CoV-2 infection in Italy and to investigate which risk factors relate to SARS-CoV-2 serum positivity.

## 2. Materials and Methods 

### 2.1. Study Design and Population 

This was a single-center, prospective observational study assessing the seroprevalence of anti-SARS-CoV-2 antibodies in IBD patients at the IBD Unit of Scientific Institute for Research, Hospitalization and Healthcare (IRCCS) Sacro Cuore-Don Calabria Hospital, Negrar di Valpolicella (Verona), Italy.

Eligible patients were those who consecutively attended our referral center between 7 December 2020 and 31 January 2021. 

Men and women with a diagnosis of IBD established at least 6 months prior, and who were undergoing biological or conventional treatment were included in the study. Treatment with thiopurines and a lack of informed consent constituted the exclusion criteria.

IBD patients were distributed into two groups:(a)patients treated with a biologic drug, including: infliximab (IFX), adalimumab (ADA), golimumab (GOL), vedolizumab (VDZ), ustekinumab (UST) or any new experimental biologic drug being used in clinical trials;(b)patients treated with oral and/or topical mesalazine.

Baseline characteristics such as age, gender, body mass index (BMI), smoking habit, presence of comorbidities and IBD-related characteristics were collected for all IBD patients. 

Concomitant treatment with oral corticosteroids such as prednisone, beclomethasone dipropionate or budesonide was also reported and analyzed for both groups of IBD patients. 

IBD was defined as UC, CD or IBDU. UC disease extension was defined, according to the Montreal Classification, in proctitis (E1), left-sided colitis (E2) and extensive colitis (E3) [20].

Montreal Classification was used to describe CD location (ileal L1, colonic L2, ileo-colonic L3, and upper gastrointestinal disease L4) and behavior (non-stricturing non-penetrating B1, stricturing B2, penetrating B3, and perianal disease) [21].

Clinical activity was defined as quiescent, mild, moderate, or severe disease according to the Partial Mayo Score and the Harvey Bradshaw Index, respectively [20,21].

A questionnaire was also submitted to IBD patients in order to investigate:-possible onset of COVID-19 related symptoms in the previous 30 days (headache, cough, sneezing, vomiting, ageusia/anosmia, fever, fatigue, dyspnea, arthromyalgia, diarrhea, conjunctivitis);-results of previous nasopharyngeal swabs (antigen or molecular tests);-flu vaccination;-any close contacts with individuals testing positive for SARS-CoV-2;-protective measures: use and type of personal protective equipment (PPE), average number of people in contact with daily [22].

In accordance with an internal protocol of the hospital, all patients had their body temperature measured before entering the building. Further to this, patients undergoing on-site infusion therapy underwent a nasopharyngeal swab for SARS-CoV-2 antigen prior to their hospital appointments. Social distancing between patients was implemented in waiting areas and during biologic drug administration.

The non-IBD control group consisted of healthcare professionals working at the Sacro Cuore-Don Calabria Hospital, who had given a blood sample for SARS-CoV-2 antibody testing in January 2021, prior to being vaccinated against COVID-19.

An Excel database was used to collect data.

### 2.2. Laboratory Analysis

Serum samples were consecutively collected and stored at −20 °C.

The concentrations of anti-SARS-CoV-2 IgM and IgG antibodies were measured by commercial automated assays by DiaSorin (DiaSorin S.p.A., Saluggia VC, Italy).

The LIAISON^®^ SARS-CoV-2 IgM is a qualitative assay for the detection of IgM antibodies against S1-RBD antigens of SARS-CoV-2. Results are expressed as an index. A value of more than 1.1 was considered a positive result. 

The LIAISON^®^ SARS-CoV-2 S1/S2 IgG, performed on the Liaison XL analyzer, is a quantitative assay that detects antibodies against two subunits of the virus spike protein (S1 and S2). 

Results of more than 15.0 AU/mL were considered to be positive [23]. According to a recent study, sensitivity and specificity of this assay were 96.7% and 95.0%, respectively [24].

All serological tests were performed at a local laboratory analysis unit. Any patients testing positive for IgM and/or IgG were invited to undertake a nasopharyngeal swab for SARS-CoV-2 mRNA in order to identify and isolate any individuals with active infection. 

### 2.3. Statistical Analysis 

Continuous variables were statistically described using the mean and standard deviation (SD).

Categorical variables were described as absolute frequency and percentage, and were compared using the Chi-squared or Fisher’s exact test. 

Univariate and multivariate analyses were carried out by logistic regression to evaluate the association between baseline characteristics or COVID-19 related risk factors and anti-SARS-CoV-2 antibodies seropositivity. 

The multivariate model was adjusted for median age, gender, BMI, smoking habit, presence of comorbidities, disease type and activity, treatment type and concomitant steroid use, close contact with SARS-CoV-2 positive cases, daily risk of exposure, and type of face mask used.

Stepwise regression by backward elimination method was used for the selection of statistically significant covariates in the multivariate analysis.

A probability value (*p*) of less than 0.05 was considered statistically significant. STATA 14.2 SE was used for calculations and analysis.

## 3. Results

### 3.1. Baseline Demographic and Disease Characteristics

Three hundred and eighty-six IBD patients were included in the study. Baseline and IBD-related characteristics with a detailed comparative analysis between the IBD groups are described in Table 1.

211 patients (54.7%) were male. The mean age was 45.4 ± 15.5 years among patients on biologics and 46.2 ± 14.2 years among patients on mesalazine. 107 patients (30.3%) had at least one comorbidity. 

Of the patients enrolled, 179 (46.4%) had UC, 200 (51.8%) had CD, and 7 (1.8%) had IBDU. 266 patients (68.9%) had quiescent disease at the time of serum collection. 

311 patients (80.6%) were undergoing treatment with biologics; most of these (99 patients, 31.8%) were on VDZ. 95 patients were on IFX (30.5%), 42 on ADA (13.5%), 4 on GOL (1.3%), 57 on UST (18.3%) and 14 (4.5%) were receiving experimental biological treatment.

The mean duration of biological treatment at the time of serum collection was 17.9 ± 15.1 months. 69 patients (17.9%) were receiving concomitant treatment with corticosteroids, most of whom were on mesalazine (*p* < 0.001).

IBD-related characteristics were not statistically different between patients on biological therapy and those on mesalazine, with the exception of UC extension and CD phenotype (*p* < 0.001 and *p* = 0.009, respectively).

The non-IBD control group consisted of 2209 healthcare workers, for which the mean age was 43.7 ± 12.5 years and 37.7% were male. 

### 3.2. COVID-19 Related Questionnaire Analysis

All IBD enrolled patients completed a COVID-19 questionnaire, the results of which are summarized in Table 2. 107 patients (27.7%) reported having had at least one COVID-19 related symptom in the previous 30 days. Amongst the total cohort of IBD patients, 234 rapid antigenic tests and 54 RT-PCR tests were reported to have been undertaken in the month prior to serum sample collection.

Regarding PPE, 383 IBD patients (99.2%) reported wearing masks daily; of these, 151 (39.4%) wore FFP2 masks. 75 patients (20.2%) reported having had close contact with SARS-CoV-2 positive individuals.

There was no statistically significant difference in exposure to COVID-19 risk factors found between the two groups of IBD patients.

### 3.3. Anti-SARS-CoV-2 Antibodies Seroprevalence Analysis 

40 of 386 IBD patients tested positive for IgM and/or IgG SARS-CoV-2 antibodies, showing an overall prevalence of 10.4%. Prevalence was not statistically different between IBD groups (10.3% versus 10.7%; *p* = 0.923). 

Six of 40 IBD patients (15%) tested positive for IgM only while 12 patients (30%) tested positive for IgM and IgG. 22 patients (55%) tested positive for IgG only. 

Among IBD patients testing positive for anti-SARS-CoV-2 antibodies, 19 (47.5%) reported having been completely asymptomatic in the preceding period. 

28 patients had previously performed a rapid antigenic test to which 7 (17.5%) tested positive. A nasopharyngeal swab for RT-PCR analysis was also previously undertaken by 23 patients of which 17 (42.5%) tested positive. 

Among the 40 seropositive patients, only one (2.5%) tested positive for SARS-CoV-2 mRNA after a nasopharyngeal swab. This patient was asymptomatic and was in receipt of a self-administered oral biological drug so had not been tested as patients receiving on-site infusions were (see Figure 1).

The prevalence rate of SARS-CoV-2 serum antibodies in non-IBD healthcare professionals was 13.0% and was not significantly higher than that of IBD patients on biological therapy (10.3%; *p* = 0.173) or in the total IBD patient population studied (10.4%; *p* = 0.145). 

### 3.4. COVID-19 Risk Factors Analysis

Details of univariate and multivariate analyses are described in Table 3.

The univariate analysis showed that close contact with a SARS-CoV-2 positive individual and using a non-FFP2 mask were significantly associated with the presence of anti-SARS-CoV-2 antibodies in the serum (*p* < 0.001 and *p* = 0.047, respectively). The same factors were confirmed to be independent predictors of seropositivity at multivariate analysis (*p* < 0.001 and *p* = 0.048, respectively).

Demographic or IBD-related characteristics were not found to be associated with seropositivity in our analysis. 

## 4. Discussion

To our knowledge, this is the first study comparing SARS-CoV-2 antibody seropositivity in IBD patients on biologics with those receiving non-biological therapy. The two groups showed a similar seroprevalence rate (10.3% and 10.7%, *p* = 0.923), which was higher than that reported by most other studies. Although our rates were higher than those of other publications, they were comparable to those of our non-IBD control group (13.0%), which is in line with other studies.

A lower prevalence rate of SARS-CoV-2 serum antibodies in IBD patients has been noted in one recent report from the United Kingdom, where seroprevalence rates were 3.0% and 7.2% in adult IBD patients from Oxford and London, respectively. Rates were significantly lower than those reported in the non-IBD healthcare professional control groups [14].

Bertè et al. tested IBD patients on biological treatment in Cagliari and Milan (Italy) and in Erlangen (Germany) and found an overall prevalence rate of 2.3% for serum IgG and 3.4% for IgA [15].

A study by Bossa et al. showed a seroprevalence rate of 2.9% in patients followed up at the IBD center in the province of Foggia (Italy) while a recent report from the University of Rome “Tor Vergata” (Italy) described a SARS-CoV-2 IgG seroprevalence rate of 1.37% [16,17].

Conversely, very high prevalence rates were found in a study carried out at a referral center in Bergamo (Italy), where 21% of 90 IBD patients on biological treatment were seropositive, with no significant difference to that of non-IBD healthcare professionals. It is important to note that as Bergamo was one of the most severely affected provinces in Italy during the first wave of the outbreak, a relatively high seroprevalence is expected in its population [18].

Finally, a study of Polish patients by Lodyga et al. has been unique in finding a higher prevalence of seropositivity in IBD patients than in non-IBD healthcare professionals (4.6–6% versus 1.6–1.1%, respectively) [19].

Inconsistency between studies could be caused by several factors.

Firstly, there are considerable differences in national and local management strategies in limiting the spread of SARS-CoV-2. At the beginning of the SARS-CoV-2 outbreak in Italy, northern regions and then the rest of Italy were subjected to a complete lockdown. As a result, there was great heterogeneity in the epidemic’s intensity across the nation. Since October 2020, the Italian government has progressively enforced new restrictive measures but another strict lockdown has not been imposed, which may have led to a longer lasting diffusion of SARS-CoV-2 and to higher incidence and prevalence rates [25].

Furthermore, all available studies referred only to the first pandemic phase and were heterogeneous in terms of study cohorts and SARS-CoV-2 antibody assays used, and in terms of antibody class investigated (IgA or IgM and/or IgG) [26]. In our study, seroprevalence analysis has included not only IgG but also IgM detection.

Almost half of patients testing positive for anti-SARS-CoV-2 antibodies did not report any symptoms indicative of previous SARS-CoV-2 infection and all but one seropositive patient had no ongoing infection. 

In addition, most seropositive patients were positive for IgG only, which may be indicative of SARS-CoV-2 infection during the first wave or shortly thereafter, considering class switching from IgM to IgG [13].

Our analysis has confirmed that treatment with a biologic drug, regardless of the type, and IBD-related characteristics are not associated with increased anti-SARS-CoV-2 antibody seroprevalence.

Our results suggest that an optimized maintenance regimen, with greater exposure to the biological drug, does not increase the risk of infection, when compared with a standard regimen. 

Our findings support those of preliminary reports from the beginning of the COVID-19 pandemic, suggesting that undertaking biological therapy is not a risk factor for the development of COVID-19 [8,9,10].

Recently, published data from the Surveillance Epidemiology of Coronavirus Under Research Exclusion (SECURE) IBD registry provided valuable reassurance that immunosuppressant drugs, in particular anti-tumor necrosis factor agents (anti-TNFs), were not associated with adverse outcomes in IBD patients who developed COVID-19 [26].

Furthermore, in our study, the two IBD patient groups investigated did not differ in numbers of asymptomatic cases, incidence of close contact with SARS-CoV-2 positive individuals or protective measures adopted. 

Our analysis shows that close contact with a SARS-CoV-2 positive individual is an important risk factor for SARS-CoV-2 seropositivity regardless of IBD treatment type. It has also shown that most IBD patients used face masks daily, and that most of these were surgical or FFP2 masks. Interestingly, the logistic regression analysis revealed a protective role of FFP2 masks in particular [27].

A large control group of healthcare professionals, not yet vaccinated and subjected to a similar infectious risk in the same study period, was included in this study. In support of other similar studies, our comparison between IBD patients and non-IBD healthcare workers showed no increased prevalence of anti-SARS-CoV-2 antibodies amongst IBD patients regardless of treatment regimen [8,9,10].

The relatively high number of IBD patients from a single IBD unit and the prospective enrollment of patients are among the strengths of this study. Moreover, we have also differentiated between patients receiving biological therapy and those who were not. 

The main limitation of this study is that the non-IBD control group was not composed of individuals from the general population. As such, further studies are needed to make valid comparisons or considerations.

In conclusion, our data suggest that the seroprevalence of anti-SARS-CoV-2 antibodies is comparable in IBD patients undergoing biological therapy and in those taking mesalazine. Both groups of IBD patients also have a seroprevalence comparable to the control group of non-IBD individuals. 

IBD-related characteristics did not increase the prevalence of anti-SARS-CoV-2 seropositivity, whereas close contact with SARS-CoV-2 positive individuals and the use of non-FFP2 masks did correlate with an increased seropositivity prevalence.

## Figures and Tables

**Figure 1 medicina-57-01048-f001:**
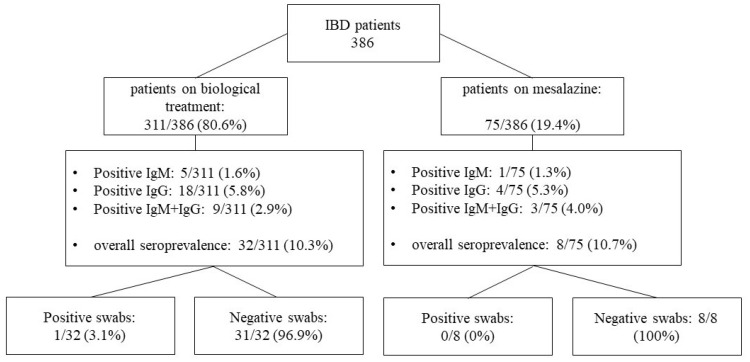
Seroprevalence analysis in IBD patients’ groups.

**Table 1 medicina-57-01048-t001:** Demographic and clinical characteristics of patients enrolled.

Characteristic	Patients on Biologic	Patients on Mesalazine	*p* Value	Overall IBD Patients	Control Group
Number of patients, *n*	311	75		386	2209
Male (%)	177 (56.9%)	34 (45.3%)	0.071	211 (54.7%)	832 (37.7%)
Mean age, years ± SD	45.4 ± 15.5	46.2 ± 14.2	0.652	45.8 ± 14.9	43.7 ± 12.5
Mean BMI, ± SD	24.8 ± 4.55	24.0 ± 4.48	0.143		
Residence:			0.911		
Veneto, *n* (%)	259 (83.3%)	62 (82.7%)			
other regions, *n* (%)	52 (16.7%)	13 (17.3%)			
Smoking habit:			0.781		
non smokers, *n* (%)	236 (75.9%)	59 (78.7%)			
ex smokers, *n* (%)	37 (11.9%)	9 (12%)			
current smokers, *n* (%)	38 (12.2%)	7 (9.3%)			
Comorbities:			0.248		
no, *n* (%)	154 (49.5%)	43 (57.3%)			
yes, *n* (%)	157 (50.5%)	32 (42.7%)			
Disease:			0.397		
UC, *n* (%)	139 (44.7%)	40 (53.3%)			
CD, *n* (%)	166 (53.4%)	34 (45.3%)			
IBD-U, *n* (%)	6 (1.9%)	1 (1.3%)			
Disease activity:			0.500		
quiescent, *n* (%)	216 (69.5%)	50 (66.7%)			
mild, *n* (%)	72 (23.2%)	17 (22,7%)			
moderate, *n* (%)	22 (7.1%)	7 (9.3%)			
severe, *n* (%)	1 (0.3%)	1 (1.3%)			
UC extension:			0.001		
E1, *n* (%)	11 (7.9%)	13 (32.5%)			
E2, *n* (%)	75 (54%)	18 (45%)			
E3, *n* (%)	53 (38.1%)	9 (22.5%)			
CD location:			0.223		
L1, *n* (%)	67 (40.4%)	19 (55.9%)			
L2, *n* (%)	32 (19.3%)	6 (17.6%)			
L3, *n* (%)	66 (39.8%)	9 (26.5%)			
L4 (upper), *n* (%)	9 (5.4%)	0 (0%)			
CD phenotype:			0.009		
B1, *n* (%)	108 (65.0%)	29 (85.3%)			
B2, *n* (%)	35 (21.1%)	4 (11.7%)			
B3, *n* (%)	22 (13.3%)	0 (0%)			
perianal disease, *n* (%)	24 (14.5%)	1 (3%)			
Biological therapy:			0.001		
IFX, *n* (%)	95 (30.5%)	0 (0%)			
ADA, *n* (%)	42 (13.5%)	0 (0%)			
GOL, *n* (%)	4 (1.3%)	0 (0%)			
VDZ, *n* (%)	99 (31.8%)	0 (0%)			
UST, *n* (%)	57(18.3%)	0 (0%)			
Others (experimental), *n* (%)	14 (4.5%)	0 (0%)			
Biological standard maintenance treatment, *n* (%)	239 (62%)	0 (0%)			
Biological optimized maintenance treament, *n* (%)	93 (24.1%)	0 (0%)			
Concomitant corticosteroid, *n* (%)	43 (13.8%)	26 (34.7%)	0.001		

Legend: SD, standard deviation; UC, ulcerative colitis; CD, Crohn’s disease; IBD-U, inflammatory bowel disease unclassified; IFX, infliximab; ADA, adalimumab; GOL, golimumab; VDZ, vedolizumab; UST, ustekinumab.

**Table 2 medicina-57-01048-t002:** COVID-19 related factors analysis.

Characteristic	Patients on Biologic	Patients on Mesalazine	*p* Value
Number of patients, *n* (%)	311	75	
COVID-19 related symptoms:			0.561
none, *n* (%)	228 (73.3%)	51 (68.0%)	
<3 simptoms, *n* (%)	60 (19.3%)	18 (24.0%)	
3–5 symptoms, *n* (%)	21 (6.8%)	5 (6.7%)	
>5 symptoms, *n* (%)	2 (0.6%)	1 (1.3%)	
Rapid antigenic tests:			0.001
positive, *n* (%)	7 (2.3%)	2 (2.7%)	
negative, *n* (%)	218 (70.1%)	7 (9.3%)	
Rt-PCR nasopharingeal swabs:			0.556
positive, *n* (%)	14 (4.5%)	5 (6.7%)	
negative, *n* (%)	30 (9.6%)	5 (6.7%)	
PPE:			0.495
surgical mask, *n* (%)	162 (52.1%)	43 (57.3%)	
FFP2 mask, *n* (%)	126 (40.5%)	25 (33.3%)	
fabric mask, *n* (%)	21 (6.8%)	6 (8.0%)	
gloves, *n* (%)	21 (6.8%)	7 (9.3%)	0.458
COVID-19 positive close contact, *n* (%)	60 (19.3%)	18 (24.0%)	0.423
Daily contacts:			0.257
only cohabitants, *n* (%):	124 (39.9%)	32 (42.7%)	
<10 individuals, *n* (%)	106 (34.1%)	18 (24%)	
>10 individuals, *n* (%)	43 (13.8%)	11 (14.7%)	
Flu vaccine taken, *n* (%)	113 (36.3%)	26 (34.7%)	0.787

Legend: rt-PCR, real-time polymerase chain reaction; PPE, personal protective equipment.

**Table 3 medicina-57-01048-t003:** Univariate and multivariate analyses of variables associated with a positive antibody seroprevalence.

Variable	Univariate Analysis		Multivariate Analysis	
	OR (95% CI)	*p* Value	OR (95% CI)	*p* Value
Age ( ≥46 vs. <46)	1.388 (0.719–2.679)	0.328		
Gender	0.879 (0.454–1.704)	0.704		
BMI (≥30 vs. <30)	1.826 (0.753–4.428)	0.182		
Smoking habit	0.370 (0.086–1.592)	0.182		
Comorbidities	1.047 (0.544–2.015)	0.890		
Disease type	0.589 (0.300–1.155)	0.124		
Disease activity	1.075 (0.534–2.165)	0.839		
Anti-TNFs	1.072 (0.436–2.634)	0.880		
VDZ	0.637 (0.220–1.843)	0.406		
UST	0.805 (0.248–2.606)	0.718		
Experimental drug	3.35 (0.849–13.208)	0.085		
Concomitant corticosteroid	0.480 (0.165–1.397)	0.178		
COVID-19 positive close contact	15.979 (7.495–34.065)	0.001	20.011 (8.746–45.784)	0.001
No-FFP2 mask	0.484 (0.229–1.023)	0.047	0.417 (0.175-0.991)	0.048
Daily risk exposure (>10 individuals)	0.866 (0.323–2.317)	0.774		

Legend: OR, odds ratio; CI, confidence interval; BMI, body mass index; Anti-TNFs, anti-tumor necrosis factor agents; VDZ, vedolizumab; UST, ustekinumab.

## Data Availability

The data presented in this study are available on request from the authors.

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
