# Peer review of "A Seroprevalence Study of Anti-SARS-CoV-2 Antibodies in Patients with Inflammatory Bowel Disease during the Second Wave of the COVID-19 Pandemic in Italy"

_medicina, 2021, doi:10.3390/medicina57101048_

Round 1

Reviewer 1 Report

This study investigated the topic issue in current setting of clinical practice for IBD. However I have some concerns.

  1. Table 3. More factors should be included in multivariate analysis.
  2. This result showed IBD status did not affect the COVID-19 condition.  I did not catch the implementation of this study well. 

Author Response

Thank you for comments and suggestions. I'll proceed with changes based on the available data. 

English language editing and revising has already been done.

Reviewer 2 Report

The research paper is well designed and contains no significant errors, and is quite interesting.

In table 1 and table 2, some variables include a P= 0.000, and all these P's should be replaced by p < 0.001.

Also, in line 162, there is a P=0.000 that should be replaced by p < 0.001. The manuscript needs to be revised to correct these errors.

In table 3, the Odds ratio of gender is written as .879. Would you please change it to 0.879? The same happens with lower confidence intervals. Would you please review all the tables?

Author Response

Thank you for comments and suggestions. I'll proceed with changes.

Paper has already been revised in English language.

Reviewer 3 Report

it is a manuscript with interest for gastroenterologists. The manuscript is well written and documented

Author Response

Thank you for comments and suggestions.

Paper has already been revised and edited in English.

Round 2

Reviewer 1 Report

I have no additional comments for this revision.